# Comparison of 17β-Estradiol Adsorption on Corn Straw- and Dewatered Sludge-Biochar in Aqueous Solutions

**DOI:** 10.3390/molecules27082567

**Published:** 2022-04-15

**Authors:** Wei Guo, Junhui Yue, Qian Zhao, Li Zhang, Shaoyong Lu

**Affiliations:** 1College of Architecture and Civil Engineering, Beijing University of Technology, Beijing 100124, China; yuejunhui3214@163.com (J.Y.); zq1176170762@163.com (Q.Z.); 2National Engineering Laboratory for Advanced Municipal Wastewater Treatment and Reuse Technology, Beijing University of Technology, Beijing 100124, China; lizhang1115@126.com; 3State Key Laboratory of Environmental Criteria and Risk Assessment, Chinese Research Academy of Environmental Sciences, Beijing 100012, China; lushy2000@163.com

**Keywords:** adsorption, 17β-estradiol, biochar, sludge, pyrolysis

## Abstract

Removal of steroid hormones from aqueous environment is of prevailing concern because of their adverse impact on organisms. Using biochar derived from biomass as adsorbent to remove pollutants has become more popular due to its low cost, effectiveness, and sustainability. This study evaluated the feasibility of applying corn straw biochar (CSB) and dewatered sludge biochar (DSB) to reduce 17β-estradiol (E2) from aquatic solutions by adsorption. The experimental results showed that the adsorption kinetics and isotherm behavior of E2 on the two biochars were well described by the pseudo-second-order (*R*^2^ > 0.93) and Langmuir models (*R*^2^ > 0.97). CSB has higher E2 adsorption capacity than DSB, and the maximum adsorption capacity was 99.8 mg/g obtained from Langmuir model at 298 K, which can be attributed to the higher surface area, porosity, and hydrophobicity of this adsorbent. Higher pH levels (>10.2) decreased the adsorption capacities of biochar for E2, while the ionic strength did not significantly affect the adsorption process. The regeneration ability of CSB was slightly better than that of DSB. The possible adsorption mechanism for E2 on biochar is suggested as π–π interactions, H–bonding, and micropores filling. These results indicated that CSB has more potential and application value than DSB on reducing E2 from aqueous solutions when considering economy and removal performance.

## 1. Introduction

Endocrine disrupting chemicals (EDCs) have received a lot of attention in recent years due to their adverse effects on the reproduction, metabolism, and growth of organisms [1,2]. Among these EDCs, the 17β-estradiol (E2) mainly derived from human and animal excretions shows the highest estrogenic activity [3], and can result in the feminization of male fishes even at the extremely low concentration of 1 ng/L [4]. Long-term exposure to E2 can result in adverse effects on the reproductive of organisms and increase cancer risk [5,6]. The recent studies confirmed that E2 in excretions can be transported to surface waters via the rainfall runoff and sewage discharge, and thus it poses potential risks to an aquatic ecosystem, such as suppressing the fish antibody-forming cell responses and aberrant expression of mRNA for estrogen receptor isoform [7,8]. Due to its higher endocrine disruptor hazard, E2 has been listed in the “the 1st Watch List under the Water Framework Directive (WFD)” by European Commission and the “Drinking Water Contaminant Candidate List 3” by the US Environmental Protection Agency (EPA) [9,10]. However, a residual concentration of E2 from few ng/L to several μg/L was still detected in various natural water systems on earth, such as lakes and rivers, due to incomplete removal during sewage and feedlots’ wastewater treatment [6,7,8,9,10,11,12,13]. Thus, the need for efficient E2 removal methods from wastewater is a crucial issue worldwide.

Up to now, several techniques have been developed to remove E2 from the aquatic environment, such as membrane filtration [14], adsorption [15], photocatalysis [16], biodegradation [17], and advanced oxidation [18]. Among them, adsorption is considered one of the most promising methods due to its low operation and maintenance costs [19]. Numerous adsorptive materials have been developed for the elimination of organic pollutants like dye and phenol from wastewaters, such as nanocomposite, resin, and neem leaves [20,21,22,23,24]. However, as the common adsorbent widely used, these adsorbents were high-cost associated with its production and relatively expensive raw materials. The potential of biochar as low-cost adsorbent for remediating steroid estrogen contamination of aqueous systems has been considered because of its large specific surface area, high porosity, and high physicochemical stability [25]. The pyrolysis temperature and the feedstock type significantly affected the adsorption characteristics of biochar with regard to E2 [26]. The maximum E2 adsorption capacity of wheat straw biochar at 298 K (62.9 mg/g) is higher than that of cow manure biochar (41.0 mg/g) [26]. The increasing pyrolysis temperature increased oxygen-containing functional groups of wheat straw biochar, which provided more adsorption sites for biochar; for example, the adsorption capacity of biochar at 550 °C was higher compared to 350 °C [27]. Moreover, the higher temperature can effectively reduce the ecological risk of biochar applications through the immobilization of heavy metals, with pyrolysis temperature of ≥600 °C being recommended [28,29].

Organic waste is often used as feedstock for the production of biochar, as it is abundantly available [25,30]. Corn straw and dewatered sludge are the most common and high-yield agricultural and municipal organic wastes from corn cultivation and sewage treatment plants, respectively, which have been reported to be used to produce biochar [28,31]. For example, the annual production of corn straw and dewatered sludge in China reached 0.26 billion tons and 6.25 million tons, respectively [32,33]. The previous studies indicated that the adsorption capacity of E2 on wheat straw biochar was higher than that on cow manure biochar, and the adsorption performance was mainly dependent on the aromaticity of biochar [26,27]. The rice husk biochar modified by microwave and alkali had a higher adsorption capacity of E2 (44.9 mg/g) [34]. The corn straw biochar addition significantly improved the adsorption rates and capacities of 17α-ethinyl estradiol on sediments [35]. However, until now, the potential application of the two low-cost agricultural and municipal organic wastes such as corn straw and dewatered sludge for E2 removal from aqueous solutions remains unclear. Therefore, the main objective of this study was to investigate the role of biochar derived from corn straw and dewatered sludge in the frame of E2 removal. Corn straw and dewatered sludge (namely CSB and DSB) were prepared at 600 °C, and characterized prior to experiments. Then, the E2 adsorption performance of the two biochars in aqueous solution was evaluated, using systematic adsorption kinetics and adsorption isotherm measurements. The influences of environmental pH and ionic strength on the adsorption process were investigated. Finally, the possible adsorption mechanisms of biochar removal E2 from water are discussed. Our study provides information related to the potential of CSB and DSB to remediate E2 from aqueous solutions.

## 2. Results and Discussion

### 2.1. Characterization of Biochar

The physico-chemical properties of CSB and DSB (i.e., elemental composition, surface area, porosity, and pH) are summarized in Table 1. CSB showed a yield (40.1%) significantly higher than that of DSB (25.9%). The ash content of CSB (9.2%) was notably lower than that of DSB (47.1%) mainly because of the decomposition of volatile substances (CO_2_) and accumulation of minerals at high content in the latter sample [36]. The carbon content of CSB was significantly higher compared to DSB, which is in accordance with the higher carbon content reported for other agricultural source biochar [37]. The O/C and H/C ratio can be used as an indication of the hydrophilicity and carbonization degree of the biochar [31]. The lower O/C ratios reflect the higher hydrophobicity [38]. The atomic O/C indicated that the order of hydrophobicity is CSB > DSB. The H/C ratio of DSB was slightly higher than that of CSB, indicating that they had similar carbonization degree and aromaticity [39]. The S_BET_ and pore volume of CSB (185.3 m^2^/g and 0.2 cm^3^/g) were 4.0 and 2.6 times greater with respect to DSB (46.3 m^2^/g and 0.06 cm^3^/g), which is consistent with the SEM results. According to SEM images (Figure 1a,b), CSB displayed a fiber layer stacking structure, while DSB showed a smooth ball stacking structure. High lignin-content biomass in corn straw easily formed more fiber structure and incompletely developed pores under pyrolysis process [31]. The average pore size of CSB and DSB were respectively determined as 26.8 and 50.1 nm, indicating that the two biochars were both mesoporous materials [26].

In the FTIR spectra images (Figure 1c), both biochars exhibited a similar population of functional groups such as −OH, aromatic C=O and C=C, C−O−C, and aromatic C−H. These functional groups of CSB showed more noticeable vibrations than that of DSB, suggesting that a significantly higher number of functional groups in CSB. Compared to CSB, DSB had some specific functional groups, for example, −CN and P=O that were respectively observed at the bands 2320 and 920 cm^−1^. The XRD patterns in the range of 10–80° of CSB and DSB are shown in Figure 1d. Obvious peaks of quartz all appeared at 2θ 26° for both CSB and DSB. Moreover, this typical peak was attributed to the amorphous structure of graphitic carbon of biochar [34]. Compared with CSB, DSB had more diffraction peaks, which proved that more inorganic crystals appear under high temperature pyrolysis, which is consistent with its high ash content (47.1%) [40]. Fewer diffraction peaks in CSB indicated its lower ash content (9.2%) and more layered graphene-like structure [31]. CSB and DSB presented both weakly alkaline (average pH values of 8.2 and 7.8, respectively). Moreover, CSB and DSB also showed low pH_pzc_ values (Table 1), thereby revealing that CSB and DSB are negatively charged at pH > 4.5 and pH > 2.7, respectively, and thus strong buffer capacity when exposed to acidic environments [41]. Compared with DSB, CSB showed the highest surface area and pore volume, and a larger number of aromatic functional groups, thereby indicating that this biochar has a higher surface adsorption potential [25].

### 2.2. Adsorption Kinetics

Adsorption kinetics can disclose crucial information about the adsorption process and the interaction mechanism of E2 and biochar. Figure 2a reveals the contact time effect of E2 adsorption onto CSB and DSB at an initial concentration of 2.0 mg/L. The kinetic curves of CSB and DSB were similar, and the adsorption capacity (Figure 2a) of E2 with the contact time increased rapidly within 4 h and then reached equilibrium after 8 h and 16 h, respectively. The equilibrium time was much longer compared to that of Fe-Mn-biochar (2 h) [42] and Mt-biochar (4 h) [26], but appeared to be comparable to that of bone char (6.5 h) [43] and sawdust biochar (15 h) [44]. Such fast adsorption process could be attributed to the hierarchically porous network, the accessibility of the abundant vacant sites, and the low internal diffusion resistance for adsorption [10]. The saturated adsorption amounts (13.8 mg/g) and removal rate at equilibrium (98.3%) of E2 for CSB was higher than that (12.5 mg/g and 89.3%, respectively) for DSB, which could be attributed to the large surface area and pore volume in CSB [45]. Moreover, there is a great correlation between the adsorption amounts of E2 and the content of hydrophobic substances of biochar [25].

To further analyze the adsorption process, three typical adsorption kinetic models (pseudo-first-order model, pseudo-second-order model, and the intraparticle diffusion model) were used to fit the experimental data (Figure 2 and Table 2) [10,46]. It appears clear that the pseudo-second-order model (*R*^2^ = 0.933–0.937) fitted the experimental data slightly better than the pseudo-first-order model (*R*^2^ = 0.912–0.927), indicating that chemisorption occurred between E2 and biochar involving valence force via sharing or exchanging electrons [34]. A three-step adsorption process was observed, based on the fitting results of the intraparticle diffusion model (Figure 2b). The first step was the film diffusion process, where E2 molecules diffused rapidly on the outer surface of biochar during the first 1 h. The second step was identified as the intraparticle diffusion process, which refers to the diffusion of E2 molecules from the biochar surface into its pores [10]. It can be seen that the slope of CSB in the second stage (2–8 h) was higher than that of DSB (2–16 h), suggesting the pore filling of E2 in CSB was more pronounced due to its greater pore volume (Table 1). The third step was the saturation of the intraparticle diffusion, where the adsorption process reached equilibrium. The fitting line did not coincide with the origin, suggesting that the adsorption was not governed solely by intraparticle diffusion and there were other rate-limiting steps [18]. As a conclusion, E2 adsorption onto CSB and DSB was dominated by the film and intraparticle diffusion together.

### 2.3. Adsorption Isotherms

The adsorption isotherms of E2 for CSB and DSB followed a similar trend. The adsorption capacity of E2 increased rapidly with the increase of the equilibrium concentrations of E2 and then slowly increased until the basic equilibrium was reached (Figure 3a). The removal rate of E2 gradually decreased with the increase of the initial concentration (from 0.5 mg/L to 4.0 mg/L) of E2. There was the highest removal rate of E2 for CSB (90.1%) and DSB (88.1%) at a low initial concentration of E2 (0.5 mg/L). This may be reason that a large number of active adsorption sites were readily accessible at lower initial concentrations [47]. The adsorption capacity and removal rate of E2 for CSB were higher than that for DSB, which may due to the higher carbon content (CSB 64.6% vs. DSB 30.9%), the larger surface area (CSB 185.3 m^2^/g vs. DSB 46.3 m^2^/g), and the higher carbonization and aromaticity degrees of CSB [31,48].

The adsorption isotherms describe the equilibrium distribution of an analyte between the adsorbent surface and the aqueous solution (or any other phase in contact with the adsorbent material). Two typical isotherm models (Langmuir and Freundlich) were used to explore the adsorption mechanisms of E2 onto CSB and DSB (Figure 3a). Figure 3a and Table 3 reveal that E2 adsorption data fit better to the Langmuir model (*R*^2^ = 0.974–0.998) compared to the Freundlich model (*R*^2^ = 0.953–0.965). Thus, CSB and DSB have homogeneous surfaces where monolayer adsorption would be dominant [34]. The *n* value in the Freundlich model can represent the heterogeneity degree of the isothermal adsorption process [25], while the *n* value of CSB was closer to 1 than that of DSB (Table 3), suggesting that the adsorption of E2 on CSB was mainly linear and the actual adsorption capacity is higher [49]. The maximum adsorption capacity of E2 on CSB and DSB estimated from Langmuir model was 99.8 mg/g and 27.0 mg/g, respectively. The adsorption of E2 on CSB was higher with respect to the activated carbon (67.6 mg/g), Mt-biochar (41.0–62.9 mg/g), graphene oxide (52.9 mg/g), and bone char (10.1 mg/g) [26,43,50,51]. In addition, CSB exhibited a shorter adsorption equilibrium time (<8 h) together with a higher raw material yield (about 6.25 million tons corn straw per year in China) [33]. Therefore, the CSB could be used as a more effective adsorbent for E2 removal from aqueous solution compared to DSB.

The Dubinin–Radushkevich (D–R) isotherm relates the heterogeneity of energies close to the adsorbent surface [52]. If a very small sub-region of the sorption surface was considered and assumed to be approximately by the Langmuir isotherm, the quantity can be related to the mean sorption energy, E (kJ/mol), which indicated the information about adsorption process as a physical or chemical ion-exchange [53]. Generally, the value of E is less than 8 kJ/mol, indicating physical adsorption, while a value of more than 8 kJ/mol suggests chemical adsorption [54]. The E values of E2 adsorbed onto CSB and DSB calculated from the D–R model plots (Figure 3b) were respectively 8.11 and 8.57 kJ/mol (Table 3), indicating that the adsorption process was driven by chemisorption [53]. This further confirmed that chemisorption of E2 is probably taking place on the surface of biochars during the process.

### 2.4. Effect of pH and Ionic Strength

The environmental pH can influence the surface charge on the adsorbent particles, as well as the ionization potential of chemicals [55]. Figure 4a shows the effect of pH ranging from 3 to 11 on the adsorption of E2 onto CSB and DSB. A significant difference is observed regarding pH influence on E2 adsorption when comparing the two biochars (ANOVA, *p* < 0.04). The E2 adsorption capacity of CSB was higher than that of DSB at all pH values. With the increase of the solution pH from 3.0 to 10.0 and values even exceeding this latter, the adsorption capacity becomes lower and equilibrium time is longer, which is in line with the results reported by Tao et al. [42]. These might be ascribed to the change in the surface charge of biochar and the speciation of E2 at different pH values [25,45]. The E2 molecules were gradually ionization and deprotonated as the solution pH increased from 7 to 11 [45]. When the pH value of solution exceeding 10.2 (p*K*_a_ of E2 is 10.2), deprotonated E2 molecules were dominant, thus the E2 molecules became negative charges [56]. When the pH of the solution is less than 10.2, the E2 molecule is mainly characterized by positive charge. According to the point of zero charge values of CSB (pH_pzc_ = 4.5) and DSB (pH_pzc_ = 2.7) (Table 1), the CSB and DSB surface were negatively charged at pH > pH_pzc_ and showed a positive charge at pH < pH_pzc_, respectively. The electrostatic attraction will occur between the positively charged E2 and the negatively charged biochars’ surface at pH_pzc_ < pH < p*K*_a_. During the adsorption process, the hydroxyl group of E2 interacts with the functional groups on the surface of biochar through H–bonding [57]. The deprotonation of the E2 molecules that is caused by the higher environmental pH (>10.2) inhibited the formation of h-bonds. In addition, electrostatic repulsion between the negative charge on the surface of biochars and the E2 anion may occur (pH > 10.2), which may slightly hinder the adsorption of E2. The effect of H-bonding on E2 adsorption onto CSB and DSB was greater compared to that of electrostatic interactions.

Furthermore, the influence of ionic strength on E2 adsorption onto CSB and DSB was investigated using NaCl solutions (0.0–1.0 M) (Figure 4b). There was a significant difference in ionic strength effect on E2 adsorption onto the two biochars (ANOVA, *p* < 0.002). The E2 adsorption capacity of CSB was higher than that of DSB at all ionic strengths. An ionic strength between 0.0 and 0.05 M did not exert any obvious influence on the E2 adsorption, while the adsorption performance displayed a slight increase using a NaCl concentration of 0.1–1.0 M, due to the salting-out effect. The decrease solubility of hydrophobic organic compounds induced by the increase of ionic strength (that is the salting-out effect), which may slightly enhance their hydrophobic interactions with biochar and was conductive to E2 adsorption [10]. Moreover, the forces of attraction between the cations and the negatively charged biochar (BC) surface are conducive for the formation of outer-sphere complexes [58]. When the concentration of Na^+^ increased, the repulsive forces between the negatively charged BC and E2 were mitigated by the cationic bridging effect caused by the complexation of BC-E2-Na or BC-Na-E2, which led to an increase in E2 adsorption, whereas Na^+^ would not have competed for the inner-sphere sites. This independence of sorption with background electrolyte concentration has been interpreted to indicate that the sorption process is primarily non-electrostatic in nature [59]. Thus, an appropriate ionic strength like NaCl in aqueous solution is conducive to the E2 adsorption.

### 2.5. Regeneration and Reusability

The regeneration and reusability of biochar is an important consideration as it relates to the operating cost and the feasibility of its practical application [44]. The related experiments were carried out for four cycles (adsorption–desorption process) under the identical experimental conditions [34]. After every cycle, CSB and DSB were collected and washed several times with ethanol and deionized water and then dried at 80 °C for next cycle use. The adsorption of recycled CSB and DSB per regeneration is shown in Figure 5. The initial adsorption capacities of CSB and DSB were 14.1 and 12.7 mg/g, respectively. After four adsorption–desorption cycles, their adsorption capacities were 12.5 and 11.0 mg/g, respectively, which decreased by 11.1% and 13.2%, respectively. The reduction of E2 adsorption amounts could be attributed to the following reasons: (1) adsorption of E2 or intermediates on the surface of biochars; (2) a reduction in the surface area and pore volume of biochars; and (3) consumption of functional groups on biochars. These results indicated that both biochars had good regeneration and reusability. Meanwhile, the regeneration ability of CSB was found to be slightly better than DSB.

### 2.6. Possible Mechanism Analysis

Through this analysis, it is observed that the adsorption of E2 onto biochar could be controlled by multiple processes. CSB and DSB obtained by the pyrolysis of corn straw and dewatered sludge at 600 °C have the highest surface area, pore volume, and aromatic functional groups, which were beneficial to E2 adsorption due to the availability of more adsorption sites. The FTIR spectrum of biochar (Figure 1c) showed that the surface of CSB and DSB produced more aromatic groups such as aromatic C=O and C=C at 1590 and 1620 cm^−1^ and aromatic C–H at 760 and 800 cm^−1^. The highly aromatic nature of biochar shows the π–electron donor and acceptor properties [60]. The E2 molecules have fused aromatic rings that are rich in π–electrons, and, consequently, the presence of π–π interactions between E2 and biochar could be speculated. In addition, the two biochars all contained oxygen-containing functional groups such as −OH at 3420 cm^−1^ and C−O−C at 1130 cm^−1^, and could be bound to the hydroxyl group of E2 via hydrogen bonds. Generally, π–π interactions occur between the phenolic moiety of E2 and the electron-accepting groups attached to the biochar surface [57], while H–bonding is formed between the phenolic and hydroxyl groups of E2 and the hydroxyl, carbonyl, and carboxyl groups of the biochar surface [61]. Therefore, π–π interactions and H–bonding may be the main adsorption mechanism of E2 onto CSB and DSB. The results of adsorption kinetics, isotherms, and pH effects also confirmed that π–π interactions and H-bonding played a significant function between biochar and E2 contaminant. The adsorption process is affected by both surface adsorption and internal pore filling. The proposed mechanism for E2 adsorption on CSB and DSB is given in Figure 6.

## 3. Materials and Methods

### 3.1. Chemicals and Materials

E2 (C_18_H_24_O_2_, molecular weight 228.29, 98% in purity) were purchased from Sigma Aldrich (St. Louis, MO, USA). Other chemicals including HCl, NaOH, NaCl, NaN_3_, and HPLC grade methanol and acetone were purchased from Beijing Chemical Co., Ltd (Beijing, China). Deionized water was produced using a Milli-Q system (Millipore Co., Molsheim, France). The stock solution (1000 mg/L) of E2 was prepared by dissolving 10 mg of E2 powder into 10 mL methanol. The solutions with different E2 concentration that were used in batch experiments were produced through the dilution of the stock solution. Corn straw and dewatered sludge samples were collected from the test field of the Beijing University of Agriculture (40.22° N, 116.23° E) and Gaobeidian sewage treatment plant in Beijing (36.68° N, 115.78° E), respectively. The corn straw was washed three times with ultrapure water to remove impurities. Dewatered sludge was directly stored at −20 °C without any pretreatment to reduce the effects of microbial degradation. Then, the two feedstocks were oven-dried at 105 °C for 24 h, crushed into powder and sieved (0.2 mm particle size), to be then eventually stored within an airtight plastic bag.

### 3.2. Preparation and Characterization of Biochar

The pretreated feedstock was pyrolyzed using a horizontal tube furnace (SK-1200, Tianjin Zhonghuan Test Electrical Furnace Co., LTD, Tianjin, China) under nitrogen (N_2_) and at a temperature of 600 °C, achieved with a heating rate of 10 °C/min. The dwell temperature was maintained for 1.5 h under N_2_ flow. Thereafter, the obtained biochar was washed with a 1 M HCl aqueous solution to remove inorgnic carbon, followed by washing with deionized water until the pH of the wash water was neutral [62]. Finally, the biochar samples were oven-dried at 105 °C for 24 h, and gently ground to allow for them to pass through a 0.15 mm sieve, eventually making the final biochar samples CSB and DSB.

The elemental composition of biochar was analyzed by means of an elemental analyzer (Vario EL, German Elementar Co., Hanau, Germany). The Brunauer–Emmett–Teller (BET) surface areas and the pore volume were determined from the N_2_ adsorption isotherm data obtained at 77 K, using a specific surface area and pore size analyser (ASAP 2020, Micromeritics, Harrisburg, PA, USA). Scanning electron microscopy (SEM, S250MK3, Cambridge UK Co., Cambridge, UK) was used to observe the surface morphology and structure of biochar. Fourier transform infrared spectroscopy (FTIR, Germany BRUKER Spectrometer Co., Karlsruhe, Germany) was used to identify the functional groups on the surface of the biochar. The X-ray diffraction (XRD) analysis of two different biochars was carried out on a Bruker D8-Advance X-ray diffractometer (Bruker, Karlsruhe, Germany) with Cu Kα radiation (Kα = 1.54 nm) at a voltage of 40 kV and a current of 40 mA. The ash content of the biochars was determined by combusting the samples in a muffle furnace at 873 K for 4 h and subsequent cooling in a desiccator until constant weight. Moreover, the zeta potential of biochar was determined using a potential analyzer (Zetasizer Nano, Malvern Panalytical, Ltd., Malvern, UK), while the pH of the biochar was measured by adding biochar to ultrapure water with a char:water ratio of 1:20.

### 3.3. Adsorption Experiments

The E2 solutions used in the adsorption experiments were obtained by diluting the E2 stock solution with deionized water containing 0.01 M NaCl and 200 mg/L NaN_3_ (used as a bio-inhibitor). According to the reported by Dropkin and Carmi [63], the rotational speed up to 150 rpm can form free convection currents in the vessel. Thus, many adsorption experimental studies used this agitation intensity to achieve adequate mixing of the adsorbate and adsorbent [27,34,42,59]. Previous studies and preliminary experiments showed that the adsorption of E2 reached equilibrium within 48 h and the optimum solid-to-liquid ratio of biochar to solution was 0.14 g/L [35,42]. Thus, in batch adsorption experiments, 5 mg of biochar were loaded into a 50 mL Teflon centrifuge tube containing 35 mL of E2 solution. All the adsorption experiments were carried out at 150 rpm in an orbital shaker for 48 h, and the temperature was controlled at 25 ± 1 °C. Every adsorption experiment (including the blanks) was run in triplicate. For adsorption kinetics analysis, samples were collected from 0.5 to 48 h at the initial E2 solutions pH value of 7 (2.0 mg/L). The experiments of adsorption isotherms were conducted by setting the initial E2 concentrations to 0.5–4.0 mg/L. The initial concentration range of E2 was set to simulate livestock breeding wastewater [42,64,65]. The influence of pH on E2 adsorption was investigated by adjusting the solution pH in the 3–11 range with 0.1 M HCl or NaOH from the initial E2 solution of 2 mg/L. The effect of ionic strength was investigated by adding various concentrations (0.001–1.0 M) of NaCl to the E2 solution. The regeneration and reusability of biochars were evaluated using four adsorption–desorption cycles experiments. After the adsorption experiments, the solids were separated from the solutions by centrifugation at 5000 rpm for 10 min, and the supernatant was filtered with a 0.22 μm Teflon filter. About 1 mL supernatant was analyzed for E2 by LC-MS (Agilent 1200 high-performance liquid chromatography, equipped with an electrospray ionization source and coupled with an Agilent 6310 ion trap mass spectrometer), which was operated in negative mode [65]. The removal efficiency and adsorption capacity (*q_e_*) were calculated based on the following equations:(1)$$Removal\;\left(\%\right)=\frac{\left(C_{0}-C_{e}\right)}{C_{0}}\times 100\%$$
(2)$$q_{e}=\left(C_{0}-C_{e}\right)\times V/m$$
where *C_0_* and *C_e_* (mg/L) represent the initial E2 concentration and the equilibrium one, *m* represents the adsorbent mass (g), and *V* represents the solution volume (L).

### 3.4. Mode of Data Analysis

Three adsorption kinetic models, namely the pseudo-first order, pseudo-second order, and intraparticle diffusion models, were used to describe the mechanism of adsorption process [34,46]:

Pseudo-first order model:*ln* (*q_e_* − *q_t_*) = *lnq_e_* − *k*_1_*t*(3)

Pseudo-second order model:*t*/*q_t_* = 1/(*k*_2_ *q_e_*^2^) + *t*/*q_e_*(4)

Intraparticle diffusion model:*q_t_* = *k_p_ t*^1/2^ + *C*(5)
where *q_t_* and *q_e_* (mg/g) are the adsorbed amount of E2 at time *t* and at equilibrium, respectively. *k*_1_ (1/min) and *k*_2_ (g/(mg min)) are the first-order and second-order apparent adsorption rate constants, respectively. *k_p_* (mg/(g min^1/2^)) represents the rate constant of the intraparticle diffusion stage, while *C* is the thickness of intraparticle diffusion kinetics species at the adsorption surface.

The Langmuir and Freundlich models were applied to fit the adsorption isotherms data and estimate adsorption coefficients [26]. The Langmuir model describes a homogenous model with uniform active sites and monolayer surface coverage but with no interactions between adsorbate molecules on neighboring sites [66]. Pollutants are adsorbed onto the surface of adsorbents with identical and homogeneously distributed sites and the number of adsorption sites is limited [67]. The Freundlich isotherm is an empirical equation assuming that the adsorption process takes place on heterogeneous surfaces and adsorption is multilayer adsorption based on the hypothesis of adsorbent surface irregularity [68]. It can predict an increase in the concentration of the ionic species adsorbed on the adsorbent surfaces with an increasing concentration of such species in the liquid phase [69]:

Langmuir isotherm model:(6)$$\frac{C_{e}}{q_{e}}=\frac{C_{e}}{q_{m}}+\frac{1}{q_{m}k_{L}}$$

Freundlich isotherm model:(7)$$lnq_{e}=nlnC_{e}+lnk_{F}$$
where *q_e_* (mg/g) and *C_e_* (mg/L) are the amount of E2 adsorbed at equilibrium and the solution equilibrium concentration, respectively. *q_m_* (mg/g) is the saturation sorption capacity of the solute, and *k_L_* (L/mg) is the Langmuir affinity coefficient. *k_F_* is the Freundlich equilibrium constant (mg/g)/(mg/L)^1/*n*^, while n is a dimensionless constant of the Freundlich equation which varies with the degree of heterogeneity of the adsorbing sites.

The D–R isotherm model is used to determine whether the adsorption process is physical or chemical [54]. The equation of the D–R isotherm model is written as follows:(8)$$lnq_{e}=lnq_{m}-k_{E}\varepsilon^{2}$$
(9)$$\varepsilon =RTln\left(1+\frac{1}{C_{e}}\right)$$
where *q_e_* is the amount of E2 adsorbed per unit dosage of adsorbent (mol/g), *q_m_* is the theoretical monolayer sorption capacity (mol/g), and *k*_E_ represents the constant of the sorption energy (mol^2^/kJ^2^), which is related to the average energy of sorption per mole of sorbate. Finally, ε is the Polanyi potential, *T* is the solution temperature (K), and *R* is the gas constant (8.314 J/(mol K)).

The value of mean sorption energy E (kJ/mol) can be calculated from the D–R parameter *k*_E_ as follows:(10)$$E=\frac{1}{\sqrt{-2k_{E}}}$$

### 3.5. Statistics

Based on the experimental results, several statistics of E2 were calculated including the mean and standard deviation (SD). Statistical analysis was conducted using the IBM SPSS Statistics software, version 18.0. The statistical differences were assessed using the one-way ANOVA test. A significance level of 0.05 was selected for all tests. The data fitting to corresponding model together with the data plotting were carried out by Origin Pro 8.5.

## 4. Conclusions

The two biochars (CSB and DSB) derived from core straw and dewatered sludge were prepared and used as a sorbent to reduce E2 pollution in aqueous environments. CSB had the higher yield (40.1%) and the lower ash content (9.2%) than that (25.9% and 47.1%, respectively) of DSB. The surface area and pore volume of CSB were significantly higher than that of DSB. The SEM image, FTIR spectrum, and XRD analysis indicated that CSB had a more layered graphene-like structure and aromatic functional groups. The adsorption experiment results confirmed that the adsorption kinetics and isotherm behavior of E2 on CSB and DSB were well described by the pseudo-second-order (*R^2^* = 0.933–0.937) and Langmuir models (*R^2^* = 0.974–0.998). Chemisorption is the main adsorption rate-limiting step, and the adsorption process was dominated by the film and intraparticle diffusion. CSB showed a higher E2 adsorption capacity than DSB. The maximum adsorption amount of E2 onto CSB and DSB at 298 K was respectively 99.8 and 27.0 mg/g based on the Langmuir isotherm model. π–π interactions, H-bonding, and pore filling are allegedly the main adsorption mechanism. The higher pH (>10.2) and ionic strength (>0.1 M) can affect the adsorption capacity of E2 onto biochar. Both biochars had a good regeneration and reuse ability, but the ability of CSB was slightly better than DSB. Therefore, the high yield and adsorption properties of CSB suggested that this biochar could be used as a more promising adsorbent for E2 removal from aqueous solution.

## Figures and Tables

**Figure 1 molecules-27-02567-f001:**
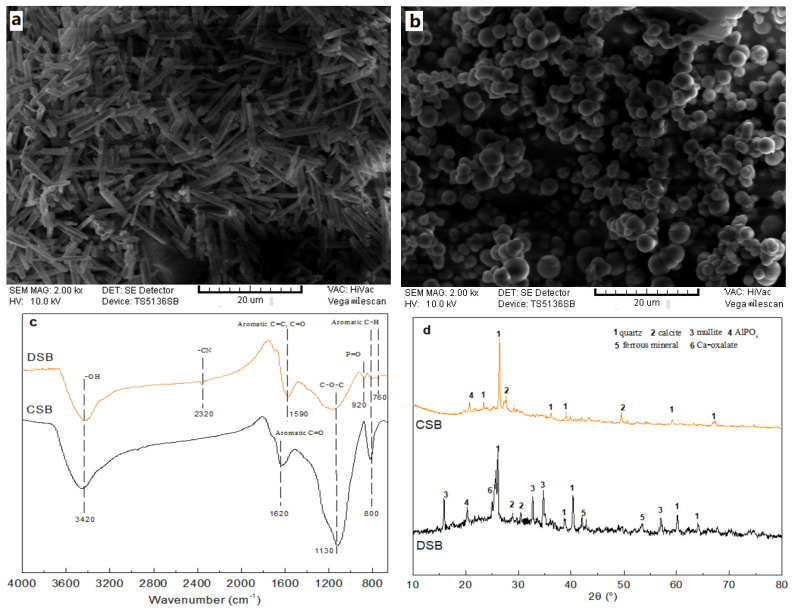
SEM images of CSB (**a**) and DSB (**b**), and the FT-IR spectrums (**c**) and XRD patterns (**d**) of the two biochars. CSB: corn straw biochar, DSB: dewatered sludge biochar.

**Figure 2 molecules-27-02567-f002:**
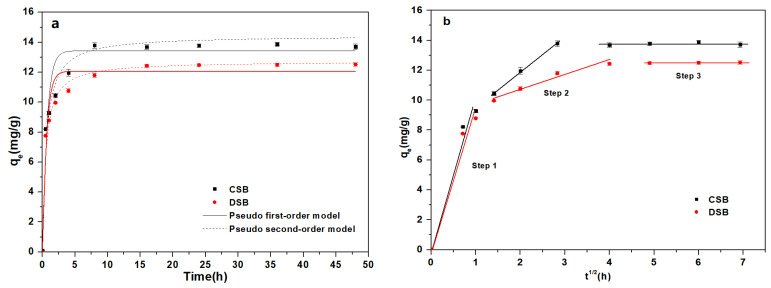
(**a**) Kinetics of 17β-estradiol (E2) onto corn straw and dewatered sludge biochars (CSB and DSB). The dotted line is the pseudo-first-order model, and the solid line is the pseudo-second-order model; (**b**) intraparticle diffusion model for the E2 adsorption on CSB and DSB.

**Figure 3 molecules-27-02567-f003:**
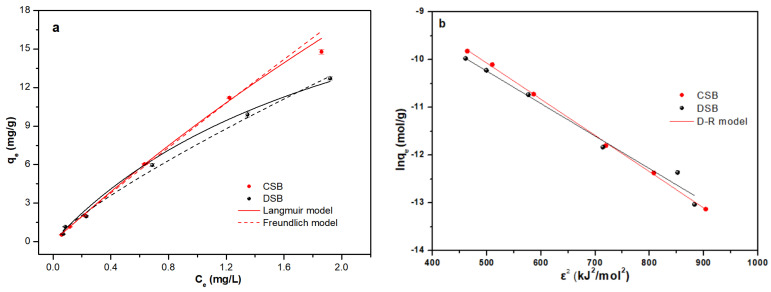
(**a**) Isotherms of 17β-estradiol (E2) adsorption onto corn straw and dewatered sludge biochars (CSB and DSB). The solid line represents the Langmuir model, and the dotted line represents the Freundlich model; (**b**) D-R isotherm plots for the E2 adsorption on CSB and DSB.

**Figure 4 molecules-27-02567-f004:**
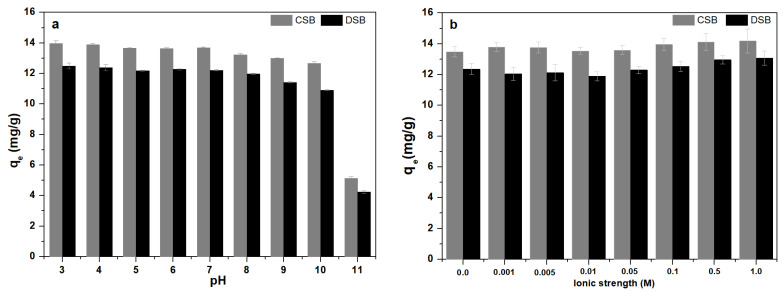
(**a**) Effect of pH and (**b**) ionic strength on 17β-estradiol (E2) adsorption onto corn straw and dewatered sludge bio chars (CSB and DSB).

**Figure 5 molecules-27-02567-f005:**
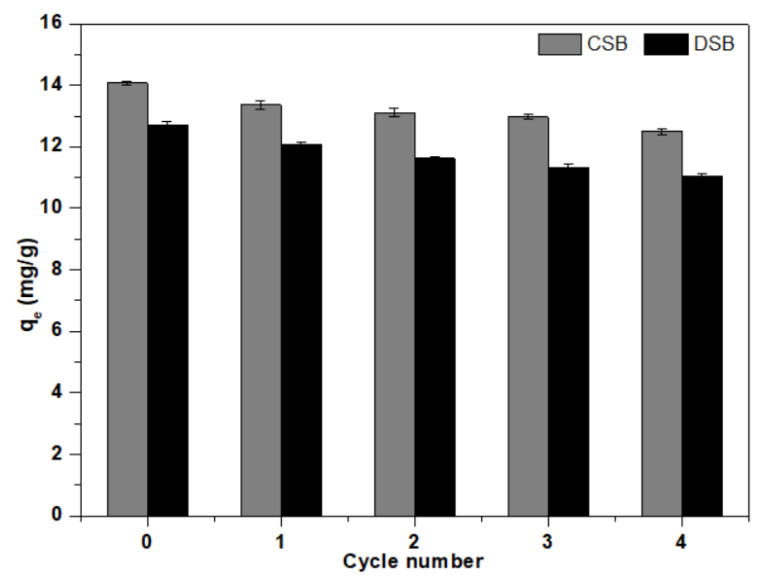
Adsorption capacity of 17β-estradiol (E2) onto corn straw and dewatered sludge biochars (CSB and DSB) during four adsorption–desorption cycles.

**Figure 6 molecules-27-02567-f006:**
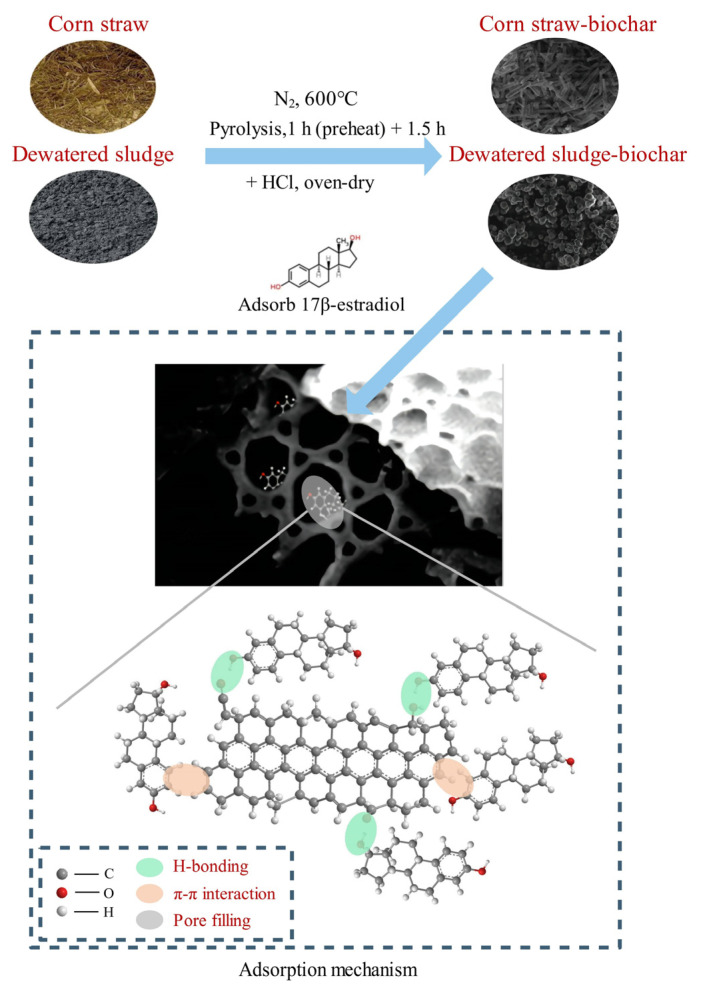
The proposed mechanisms for E2 adsorption on biochars.

**Table 1 molecules-27-02567-t001:** Physical and chemical properties of corn straw and dewatered sludge biochars (CSB and DSB).

Absorbent	C	H	N	O	H/C	O/C	Ash Content	Yield	S_BET_ ^1^	Pore Volume	Pore Size	pH_pzc_	pH
	%	%	%	%	%	%	%	%	m^2^/g	cm^3^/g	nm		
CSB	64.6	2.6	2.4	14.1	0.5	0.2	9.2	40.1	185.3	0.2	26.8	4.5	8.2
DSB	30.9	1.6	4.3	12.4	0.6	0.3	47.1	25.9	46.3	0.06	52.1	2.7	7.8

^1^ S_BET_ represents the specific surface area.

**Table 2 molecules-27-02567-t002:** Adsorption kinetic parameters of 17β-estradiol (E2) on corn straw and dewatered sludge biochars (CSB and DSB).

Kinetic Models	Parameter	CSB	DSB
Pseudo-first order	*q*_e_ (mg/g)	13.4	11.9
	*k*_1_ (1/min)	0.03	0.02
	*R* ^2^	0.927	0.912
Pseudo-second order	*q*_e_ (mg/g)	14.4	12.7
	*k*_2_ (g/mg min)	0.002	0.003
	*R* ^2^	0.937	0.933

**Table 3 molecules-27-02567-t003:** The adsorption isotherm parameters of 17β-estradiol (E2) by corn straw and dewatered sludge biochars (CSB and DSB).

Absorbent	Langmuir Model			Freundlich Model			D–R Model	
	*q* _m_	*k* _L_	*R* ^2^	*k* _F_	*n*	*R* ^2^	E	*R* ^2^
	mg/g	L/mg		((mg/g)/(mg/L)^1*/n*^)			kJ/mol	
CSB	99.8	0.11	0.998	9.07	0.95	0.965	8.11	0.978
DSB	27.0	0.45	0.974	7.58	0.81	0.953	8.57	0.949

## Data Availability

Not applicable.

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
