# Peer review of "Comparison of 17β-Estradiol Adsorption on Corn Straw- and Dewatered Sludge-Biochar in Aqueous Solutions"

_molecules, 2022, doi:10.3390/molecules27082567_

Round 1

Reviewer 1 Report

Comments on molecules-1627577:

  1. Lines 32-34: include other detrimental effects of E2 towards humans and the environment. Discuss in one sentence also its mobility in the aqueous environment.
  2. Lines 57-61: include what are sources of core straw and dewatered sludge.
  3. Lines 61-63: explain clearly what previous studies have done this and what is the source of their biochar.
  4. Lines 83-86: any preparative method for dewatered sludge?
  5. Line 110-112: why was 5 mg biochar selected as adsorbent weight? 150 rpm as agitation rate? 48 h as contact time?
  6. Line 116: why select initial E2 concentrations to be within the range of 0.5 - 4.0 mg/L?
  7. Lines 139-140: include assumptions behind Langmuir and Freundlich. Indicate also the corresponding references where these equations have originated.
  8. 4 and 5: provide the linear form since Eq. 6 is in linear form.
  9. Lines 159-160: if E > 16 kJ/mol, what would it imply?
  10. Lines 178-179: SEM of CSB showed smoother surface when compared to DSB with a rougher morphology, which does not agree with the BET results where CSB has higher pore volume and surface area.
  11. Line 187: provide exact value of pHpzc of both CSB and DSB.
  12. Lines 209-211: include discussion on the time to reach equilibrium for both CSB and DSB.
  13. 3a – include the theoretical curve represented by D-R isotherm as well for comparison purposes.
  14. Lines 271-281: discuss the charge of E2 in relation to the surface charges of CSB and DSB under various pH.
  15. Lines 283-291: what does the results of effect of ionic strength imply? Does it mean inner or outer sphere complexes are forming?
  16. Lines 304-310: what characterization analysis supports this discussion that π-π interactions and H-bonding were the 311 major adsorption mechanisms of E2 onto CSB and DSB?

Author Response

Reviewer #1:

Comment 1:

Lines 32-34: include other detrimental effects of E2 towards humans and the environment. Discuss in one sentence also its mobility in the aqueous environment.

Answer: The related description was added in the revised manuscript (lines 35-39 on page 1).

Comment 2:

Lines 57-61: include what are sources of core straw and dewatered sludge.

Answer: According to your suggestion, the sources description was added in the revised manuscript (lines 68-69 on page 2).

Comment 3:

Lines 61-63: explain clearly what previous studies have done this and what is the source of their biochar.

Answer: According to your suggestion, the explain were modified in the revised manuscript (lines 71-79 on page 2).

Comment 4:

Lines 83-86: any preparative method for dewatered sludge?

Answer: The method was added in the revised manuscript (lines 349-351 on page 10).

Comment 5:

Line 110-112: why was 5 mg biochar selected as adsorbent weight? 150 rpm as agitation rate? 48 h as contact time? Line 116: why select initial E2 concentrations to be within the range of 0.5 - 4.0 mg/L?

Answer: These parameters were from the previous study and experiment. The modification was in lines 381-384 on page 11.

Comment 6:

Lines 139-140: include assumptions behind Langmuir and Freundlich. Indicate also the corresponding references where these equations have originated.

Answer: According to your suggestion, this part was modified in the revised manuscript in lines 421-425 on page 12.

Comment 7:

4 and 5: provide the linear form since Eq. 6 is in linear form.

Answer: The two equations were modified in the revised manuscript in lines 427 and 429 on page 12.

Comment 8:

Lines 159-160: if E > 16 kJ/mol, what would it imply?

Answer: We checked the related literatures, the limit value of 16 kJ/mol was not correct. The related sentence was modified to “Generally, the value of E is less than 8 kJ/mol indicating physical adsorption, while a value more than 8 kJ/mol suggests chemical adsorption [54].” in lines 234-236 on page 6 in the revised manuscript.

Comment 9:

Lines 178-179: SEM of CSB showed smoother surface when compared to DSB with a rougher morphology, which does not agree with the BET results where CSB has higher pore volume and surface area.

Answer: The SEM images of CSB and DSB were remeasured. The new SEM images was in the revised Figure 1 a,b in page 4 in in the revised manuscript. And the related description was modified in lines 104-109 on page 3 in the revised manuscript.

Comment 10:

Line 187: provide exact value of pHpzc of both CSB and DSB.

Answer: We have done that in line 123 on page 3 in revised manuscript.

Comment 11:

Lines 209-211: include discussion on the time to reach equilibrium for both CSB and DSB.

Answer: This was added and modified in the revised manuscript (lines 135-141 on page 3 and 142-147 on page 4).

Comment 12:

3a – include the theoretical curve represented by D-R isotherm as well for comparison purposes.

Answer: The D-R model is inconsistent with the other two models (Langmuir and Freundlich models) in terms of independent variable functions, and is not suitable for comparison on one coordinate axis. The Langmuir and Freundlich models mainly reflect the adsorption capacity change under isothermal conditions, while the D-R model reflects the chemical or physical characteristics of adsorption, which has a certain linear relationship with the square of the adsorption energy (ε2).

Comment 13:

Lines 271-281: discuss the charge of E2 in relation to the surface charges of CSB and DSB under various pH.

Answer: This was modified in the revised manuscript (lines 259-272 on page 7).

Comment 14:

Lines 283-291: what does the results of effect of ionic strength imply? Does it mean inner or outer sphere complexes are forming?

Answer: The results of effect of ionic strength was modified in the revised manuscript (lines 280-291 on page 7).

Comment 15:

Lines 304-310: what characterization analysis supports this discussion that π-π interactions and H-bonding were the 311 major adsorption mechanisms of E2 onto CSB and DSB?

Answer: This part of the discussion was modified in the revised manuscript (lines 319-335 on page 9).

Reviewer 2 Report

In this manuscript, the authors have reported the feasibility of applying corn straw biochar (CSB) and dewatered sludge biochar (DSB) to reduce 17β-estradiol (E2) from aquatic solutions by adsorption. The experimental results showed that the adsorption kinetics and isotherm behavior of E2 on the two biochars were well described by the pseudo-second-order (R2 > 0.99) and Langmuir models (R2 > 0.97). CSB has higher E2 adsorption capacity than DSB, and the maximum adsorption capacity was 99.8 mg/g obtained from Langmuir model at 298 K, which can be attributed to the higher surface area, porosity and hydrophobicity of this adsorbent. Higher pH levels (> 10) decreased the adsorption capacities of biochar for E2, while the ionic strength did not significantly affect adsorption process. The possible adsorption mechanism for E2 on biochar is suggested as π-π interactions, H-bonding and micropores filling. These results indicated that CSB has more potential and application value than DSB on reducing E2 from aqueous solutions when considering economy and removal performance.

Overall the work is interesting and some important results are also reported. However, the authors should perform a major revision of the manuscript considering the points given below so as to consider it for publication in Molecules.

  • There is a lack of discussion on advantages of corn straw and dewatered sludge biochar over other adsorbent for E2 adsorption in the introduction section.
  • Discussion required for the effect of dose on removal percentage of 17β-Estradiol or E2 using different adsorbents (e.g., CSB and DSB).
  • Discussion required for the effect of initial concentration of 17β-Estradiol or E2 on its removal percentage using different adsorbents (e.g., CSB and DSB).
  • Discussion required for the effect of contact time on removal percentage of 17β-Estradiol or E2 using different adsorbents (e.g., CSB and DSB).
  • The results obtained from one-way ANOVA test has not been shown. Please add the same.
  • The authors should have presented the X-ray diffraction pattern (XRD) of the adsorbents to check the crystalline nature of the adsorbents.
  • To provide a clear back ground on the application adsorption technology in other areas authors may cite the following references in the introduction portion of the manuscript.

 https://doi.org/10.1080/03067319.2021.1946683;
https://doi.org/10.2166/wst.2021.506;
https://doi.org/10.2166/wpt.2021.064;
https://doi.org/10.1016/j.scp.2021.100514
https://doi.org/10.1080/01932691.2020.1845958

  • The image quality of figure 5 needs to be improved.
  • Regeneration study has not been performed, which is very important.
  • Performance evaluation study or comparison with other literatures can be provided to compare its performance with reference to adsorption capacity and other experimental conditions.
  • Some more important results obtained can be added in the conclusion part for better representation.

Author Response

Reviewer #2:

Comment 1:

There is a lack of discussion on advantages of corn straw and dewatered sludge biochar over other adsorbent for E2 adsorption in the introduction section.

Answer: The related description was added in the revised manuscript (lines 71-79 on page 2).

Comment 2:

Discussion required for the effect of dose on removal percentage of 17β-Estradiol or E2 using different adsorbents (e.g., CSB and DSB).

Discussion required for the effect of initial concentration of 17β-Estradiol or E2 on its removal percentage using different adsorbents (e.g., CSB and DSB).

Discussion required for the effect of contact time on removal percentage of 17β-Estradiol or E2 using different adsorbents (e.g., CSB and DSB).

Answer: The solid-to-liquid ratio of biochar to solution and intial E2 concentration were determined and adjusted using the results of previous experimen and the information reported by Li et al. (2020, respectively. The related description was added in the revised manuscript (lines 381-384 on page 11). The discussion of contact time on removal percentage of 17β-Estradiol or E2 using different adsorbents (e.g., CSB and DSB) was added in lines 142-145 on page 4 in the revised manuscript.

Comment 3:

The results obtained from one-way ANOVA test has not been shown. Please add the same.

Answer: The related content was added in the revised manuscript (lines 252-254 and lines 275-276 on page 7).

Comment 4:

The authors should have presented the X-ray diffraction pattern (XRD) of the adsorbents to check the crystalline nature of the adsorbents.

Answer: The results of XRD was added in the revised manuscript (lines 115-121 on page 3 and revised Figure 1d).

Comment 5:

To provide a clear back ground on the application adsorption technology in other areas authors may cite the following references in the introduction portion of the manuscript.

 https://doi.org/10.1080/03067319.2021.1946683;

https://doi.org/10.2166/wst.2021.506;

https://doi.org/10.2166/wpt.2021.064;

https://doi.org/10.1016/j.scp.2021.100514

https://doi.org/10.1080/01932691.2020.1845958

Answer: The description of adsorption technology was modified in the revised manuscript by added these literatures (lines 50-54 on page 2).

Comment 6:

The image quality of figure 5 needs to be improved.

Answer: We have done that in revised Figure 6.

Comment 7:

Regeneration study has not been performed, which is very important.

Answer: Thank for the suggestion, the relation was added in the section of 2.5 Regeneration and reusability in lines 296-310 on page 8 and in Figure 5 in revised manuscript.

Comment 8:

Performance evaluation study or comparison with other literatures can be provided to compare its performance with reference to adsorption capacity and other experimental conditions.

Answer: The discussion of the adsorption performance was modified in lines 222-225 on page 6 in the revised manuscript.

Comment 9:

Some more important results obtained can be added in the conclusion part for better representation.

Answer: Thank for the suggestion, the conclusion was rewritten in lines 456-473 on page 13 in revised manuscript.

Reviewer 3 Report

Dear Authors, it is true that the removal of steroid hormones from an aqueous environment is of prevailing concern because of their adverse impact on organisms. But:

  • using biochar derived from biomass adsorbent one can expect good adsorption as carbons are nonspecific adsorbents. Corn straw biochar (CSB) and dewatered sludge biochar (DSB) can not be exceptional, it would be surprising if they show low adsorptibility. Thus the question: what about novelty?
  • Your experimental results of the adsorption kinetics and isotherm need rest analysis!! Simple R2 is not good here. You will see that the models are not good.
  •  
  • What would be interesting, is the explanation of the fact that besides the maximum adsorption capacity calculated as high as 99.8 mg/g (obtained from Langmuir model) the real long-long term ads. capacity will be much higher!!!

Detailed problems:

  • Materials characterization is unacceptable:
    • EA: what about oxygen?
    • What about ashes? Is 1M HCl enough to remove all of them?
    • IR - horror: you cannot describe the diamond signals!!!!
    • Raman??
    • You must show XRD pattern!
    • SEM is typical but EDX seems to be necessary
  • LL 185-188: “CSB and DSB were both weakly alkaline (average pH values of 8.2 and 7.8, respectively). Moreover, CSB and DSB also showed low pHpzc values (<5.0), thereby revealing high acidic properties” J - no comment; you should decide acids or base??
  • Fig 2b – 2 points line??
  • Figs 2 & 3 rest analyses show that these are not good models

Etc. etc.

Summing, this is not good work, sorry. Lots of work should be done here to correct it. Based on the lack of novelty and errors I cannot recommend this work for publication.

Author Response

Reviewer #3:

Comment 1:

EA: what about oxygen? What about ashes?.

Answer: The related content was added in Table 1 and the related discussion was added in lines 93-95 on page 2, lines 97-100 and 119-121 on page 3 in the revised manuscript.

Comment 2:

IR - horror: you cannot describe the diamond signals!!!!

Raman??

You must show XRD pattern!

SEM is typical but EDX seems to be necessary

Answer: The FTIR and SEM were re-analyzed, and the related modification in lines 104-115 on page 3 and revised Figure 1 a,b,c. XRD pattern was added in lines 115-121 on page 3 in the revised manuscript.

Comment 3:

Is 1M HCl enough to remove all of them?

Answer: The obtained biochar was washed with a 1 M HCl aqueous solution to remove inorgnic carbon, followed by washing with deionized water until the pH of the wash water was neutral [62]. The ash content was added in Table 1. This concentration can eliminate the inorganic carbon in the biochar. Each experimental operation can be to take a small amount of biochar, and artificially remove the inorganic carbon when the bubbles are not generated, and then rinse it with deionized water and then dry it.

Comment 4:

LL 185-188: “CSB and DSB were both weakly alkaline (average pH values of 8.2 and 7.8, respectively). Moreover, CSB and DSB also showed low pHpzc values (<5.0), thereby revealing high acidic properties” J - no comment; you should decide acids or base??

Answer: Thank for the suggestion, the description was modified in lines 123-125 on page 3 in the revised manuscript.

Comment 5:

Fig 2b – 2 points line??

Figs 2 & 3 rest analyses show that these are not good models

Answer: We have redrawn Figure 2 due to problems with the previous data. We also checked the fit of Figure 3. The related modification was in lines 177-179, 182-188 on page 4 and revised Figure 2, Table 2 in the revised manuscript.

Round 2

Author Response

Comment 1:

Pg. 1, lines 37-39: specify what are these potential risks towards aquatic ecosystems. Do not be vague.

Answer: The detailed description was added in the revised manuscript (lines 37-40 on page 1).

Comment 2:

Pg. 11, lines 381-384: did not answer my question - why was 5 mg biochar selected as adsorbent weight? 150 rpm as agitation rate? 48 h as contact time? why select initial E2 concentrations to be within the range of 0.5 - 4.0 mg/L? what is the basis of the previous study?

Answer: According to your suggestion, the related description was added in the revised manuscript (lines 381-386 on page10 and lines 393-394 on page 11).

Comment 3:

Pg. 12, lines 421-425: add other assumptions for Langmuir and Freundlich. Your assumptions are too simplistic.

Answer: According to your suggestion, the assumptions were modified in the revised manuscript (lines 426-435 on page 11).

Comment 4:

Pg. 7, lines 280-291: authors did not answer my question. Fig. 4b – results show inner or outer sphere complexes are forming.

Answer: Fig. 4b showed the outer sphere complexes were formed. References were replaced to better discuss this result. The modification was in the revised manuscript (lines 284-292 on page 7).

Reviewer 2 Report

The manuscript can be accepted in its present form

Author Response

None

Reviewer 3 Report

authors ignored my most important remarks

Author Response

The manuscript was further improved.